

# In-situ Calibration of Offsetting Magnetometer Feedback Transients on the Cassiope Spacecraft

David M Miles[1], Andrew D. Howarth[2], Greg A. Enno[2]

[1]Department of Physics and Astronomy, University of Iowa, Iowa City, Iowa, 52242, USA

[2]Department of Physics and Astronomy, University of Calgary, Calgary, Alberta, T2N 1N4, Canada

*Correspondence to*: David M Miles (david-miles@uiowa.edu)

**Abstract.** We present an in-situ calibration process to derive the transient behaviour of the offsetting fluxgate magnetometer (MGF) instruments on the Cassiope spacecraft. The dynamic behaviour of the MGF changed on-orbit following a software update.

Characterising the new instrument dynamics during normal spacecraft operations and then removing the transients was confounded by significant magnetic interference from the reaction wheels used to orient the spacecraft. Special operations were performed where data was taken in a safe-hold mode, with the reaction wheels stopped, following a single-event upset of the spacecraft bus flight computer after transiting the South Atlantic Anomaly. The slow single-axis rotation of the safe-hold mode was used to characterize the fluxgate's new feedback dynamics. This characterisation process was then adapted for routine operation intervals

with slow reaction wheel rates to allow the transient behaviour to be characterized over long intervals of data spanning a wide range of temperatures. Subtracting these characterized transients from the flight data improves the instrument's noise floor and allows the instrument to accurately track rapidly changing local fields without loss of measurement fidelity. More generally, this characterisation process should apply to other situations where the dynamics of an offsetting instrument must be calibrated in-situ.

## 1 Introduction and Motivation

The Enhanced Polar Outflow Probe (e-POP) payload on-board the Cassiope spacecraft (Yau and James, 2015) includes the Magnetic Field Instrument (MGF) to study small-scale field-aligned currents (Wallis et al., 2015). The MGF comprises two matched fluxgate magnetometers, referred to as Inboard and Outboard, deployed at different distances from the spacecraft on a common boom. The MGF uses an offsetting design, where digitally controlled magnetic feedback extends the magnetic range of the instrument allowing it to maintain fine (62.5 pT) resolution even in the strong ambient field at a perigee of ~325 km. An MGF

firmware patch to address a timing issue in the instrument was required after the launch of the spacecraft. This firmware patch, described below, changed the dynamics of the instrument as it updates its magnetic feedback to track the local field. Without appropriate compensation in post-processing, the magnetic feedback updates lead to non-physical transients on the order of 50 nT in the measured magnetic field. Figure 1a shows how the processed magnetic field data can be contaminated with broadband noise, manifesting as vertical stripes in the dynamic spectra, resulting from the uncompensated transients after each update to the magnetic

feedback used to null the magnetic field in the sensor (Figure 1c).

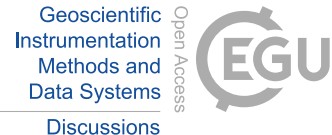


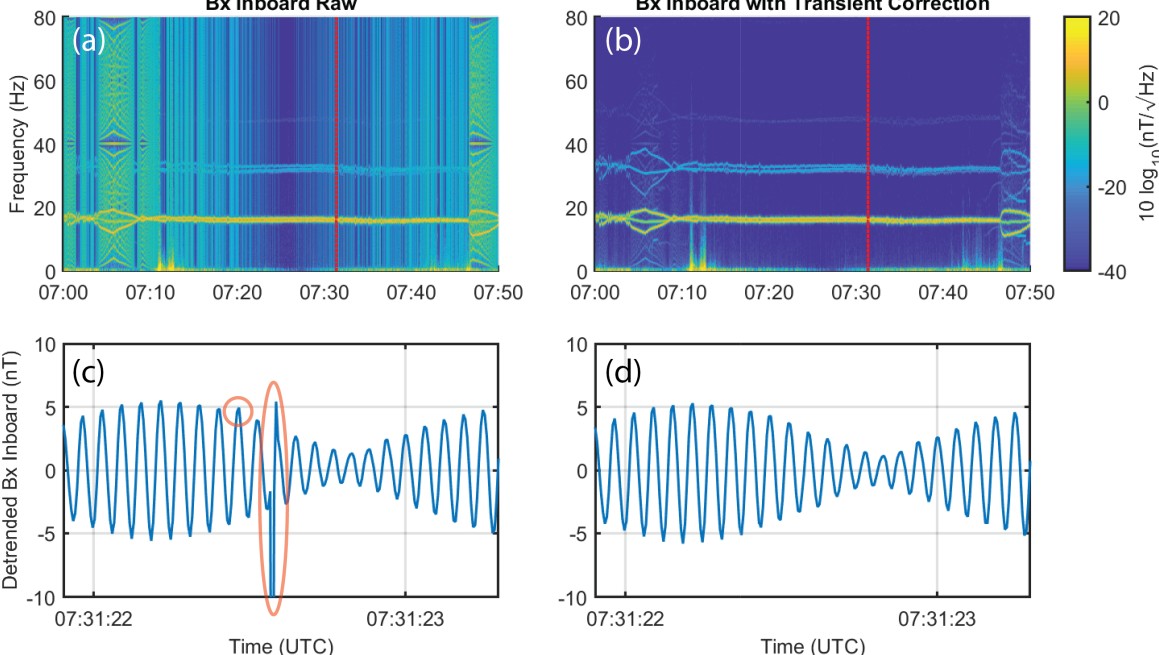

**Figure 1: Dynamic spectra of one instrument channel with (a) uncompensated transients and (b) compensated transients (cf. Figures 5-7 below). Bursts of broadband noise manifest as vertical lines in the uncompensated spectra. The magnetic signatures of the spacecraft reaction wheels are visible near 15 Hz and broaden during spacecraft manoeuvres around 07:05 and 07:50 UTC. The broadband noise results from transients after updates to the instrument's magnetic feedback shown circled in (c) and compensated in (d). The timeseries in (c) and (d) correspond to the interval in the red dashed lines in (a) and (b). The orange circles show two uncompensated feedback transients.**

Until now, these instrument transients have been mitigated by invalidating five samples after each update in post-processing and then restoring these values by interpolation. Unfortunately, during geophysically interesting intervals the local field can vary rapidly, necessitating frequent updates to the instrument's magnetic feedback and result in a high percentage of invalidated data. This can make the data interpolation poorly constrained as unaffected measurements of the magnetic field become sparse (see the example in Figure 8 below). Consequently, the MGF data can be significantly degraded during times of large magnetic fluctuations that are associated with its nominal science goal of characterising intense, small-scale field-aligned currents.

It is relatively straightforward to characterize this behaviour in a laboratory environment using a magnetic shield and then subtract the known transients from the measured data. However, after launch, the characterization was complicated by the magnetic interference from the reaction wheels used to three-axis stabilise the spacecraft. The attitude control system attempts to spin the reaction wheels at a common nominal speed, creating a complex superposition of similar frequency sinusoids which separate during spacecraft manoeuvres (Figure 1a). We present in-situ characterization of these feedback transients in the MGF instruments and their successful compensation. This data correction resulted in significantly improved noise floor (Figure 1b) and a time series reconstruction which is robust even in a rapidly varying magnetic field (Figure 1d and Figure 8 below).

## 2 Consequences of an Offsetting Magnetometer Design

The MGF is an initial step in adapting a terrestrial fluxgate magnetometer design (Narod and Bennest, 1990) for use in a space application (Miles et al., 2013, 2017; Wallis et al., 2015). The MGF is a classic second harmonic analog fluxgate (e.g., Primdahl, 1979) with the range extended by the application of a variable magnetic offset to the sensor (Figure 2). The output of a digital-to-





analog converter (DAC) is converted into a temperature-compensated current (Acuña et al., 1978; Miles et al., 2017; Primdahl, 1970) to cancel the majority of the ambient field in the sensor. This allows the forward gain of the instrument to be increased providing 62.5 pT resolution using a 12-bit analog-to-digital converter (ADC). The local magnetic field is then reconstructed as the scaled sum of the applied magnetic offset and the measured magnetic residual in the sensor.

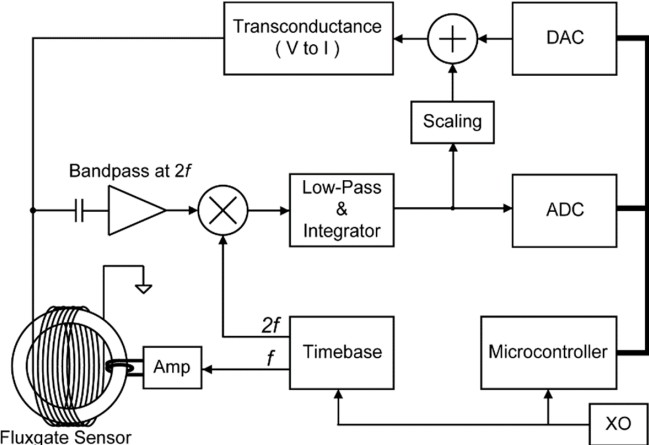

**Figure 2: Block diagram of the MGF instrument showing the offsetting design whereby a digital-to-analog (DAC) converter applies a digitally controlled offset current to the sensor to partially cancel the ambient magnetic field. Reproduced from Wallis et al., (2015).**

This offsetting design allows the instrument to preserve its resolution even in the near-Earth magnetic field at ~325 km altitude (orbit perigee) rather than having to enter a lower resolution mode, as is the case in gain-ranging instruments such as the Electric

and Magnetic Field Instrument Suite and Integrated Science (EMFISIS) on the Radiation Belt Storm Probes (RBSP) mission (Kletzing et al., 2013). However, this offsetting design poses challenges in space applications as the instrument must continuously servo the magnetic offset in each of the three axes to correctly track the continuously changing field in the frame of the sensor as the spacecraft orbits the Earth. Updates to the DAC take a finite amount of time to propagate through the transconductance circuit and the amount of magnetic feedback experienced by the sensor varies while the DAC and filter are settling, creating non-physical

transients in the reconstructed measurement of the magnetic field.

This transient behaviour is typically characterized and removed following each DAC update to reconstruct an accurate measurement of the local magnetic field. However, a bug in the as-launched MGF firmware caused variable timing between the updates to the DAC and the ADC sampling of the residual field in each magnetometer channel. Consequently, the ADC sampled an arbitrary phase of the settling filter response and the transient in the reconstructed magnetic measurement varied and could not

be characterized and subtracted.

This behaviour was verified in the laboratory by placing the engineering spare sensor in a magnetic shield and applying a slowly varying magnetic ramp (Figure 3a). The transients after each DAC update create the ticks visible on the ramped magnetic field. Figure 3b shows sequences of measurements, adjusted to zero before a DAC update, showing the envelope of possible transients caused by sampling the settling filter after a non-constant delay. New instrument firmware (V1.3.0) was developed to make the

offset between the DAC update to ADC sampling constant; this stabilised the dynamic behaviour of the DAC updates (Figure 3c). The two MGF flight instruments received the new firmware in April 2014. However, processing the data generated by the new firmware was complicated by the transient behaviour after a DAC update being unique to each individual sensor and electronics hardware combination. The transient behaviour of the engineering spare hardware, which was simple to characterise in the laboratory, could not be directly applied to the updated flight hardware on-orbit.



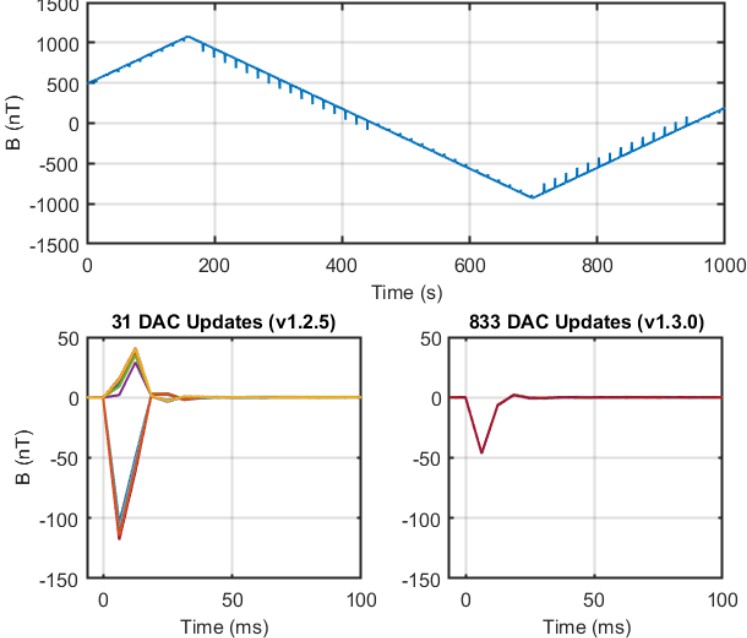

**Figure 3: (a) Ramped test magnetic field applied in the laboratory to demonstrate the transient behaviour of the engineering spare magnetometer. The ticks in the measured field are uncompensated feedback transients. (b) Zoomed and de-trended view of over plotted transients observed in the output of the sensor due to the original variability timing of the digital magnetic feedback. (c) Same but with the firmware updated to provide repeatable timing.**

The challenge was to characterize the transient behaviour on each axis of the inboard and outboard flight sensors sufficiently to allow for compensation of the transients in the post-processed flight data. This was further complicated by the on-orbit data being contaminated by 5 to 25 nT of local magnetic noise from the reaction wheels.

### 3 In-Flight Calibration

The laboratory characterization technique cannot be directly applied to in-situ data. The reaction wheels used to orient the spacecraft create a complex ~10-15 Hz local field (as-launched, wheels rates were changed in 2016 as described below) that varies on the same time scale as the transients; filtering to remove the sinusoidal signal created by the reaction wheels affects the impulse response of the system and modifies the transient behaviour resulting in an incorrect correction.

For mission reliability reasons, the wheels are not permitted to be commanded off during normal operations as there is a stiction failure mode that had been observed in prior missions. Serendipitously, the on-board spacecraft computer has experienced a little over a dozen reboots since launch, many of these occurring over the South Atlantic Anomaly. As part of the recovery, the spacecraft enters a safe-hold mode where it automatically shuts down the reaction wheels. In this mode, the magnetorquers orient the main solar panel to the sun and trigger a ~400 s spin around the instrument Z axis to minimise differential heating due to sunlight (Figure 4, top). Normally, no science data is acquired in this mode. However, the spacecraft operations team developed a special recovery process where the MGF was operated for ~30 minute observing sessions before the spacecraft was fully restored into normal operations, which included powering back on the reaction wheels.

Two of these 30-minute no-wheel intervals were used for an initial fit of the transient behaviour. The calibration was undertaken by looking for DAC updates on each channel that were separated by at least 32 samples to ensure that the analog electronics had



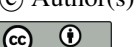

fully settled before they were perturbed by the subsequent DAC update. Further, the transients weakly couple between the X, Y, and Z axes of each sensor so intervals were only used if the other two channels had no updates during the period to prevent the characterisation of each channel being contaminated by updates on the other two channels. The primary spacecraft motion was rotation around the Z axis (Figure 4a) which provided ~2000 usable DAC updates in X and Y but only ~10 in Z.

5    The sinusoidal trend created by the spacecraft spin can be approximated as linear over the 32 sample (200 ms) interval allowing the samples following a DAC update to be estimated and the transient tick to be measured. Robust linear regression was used to fit and remove any background trend during the interval (Figure 4b). The known scaling between the feedback from the DAC (32 nT/bit) and the ADC forward loop (0.0625 nT/bit) was used to subtract the step function expected after a DAC update and reveal the transient behaviour of that axis (Figure 4c). Robust linear regression was used again to ensure that the transient starts and ends

10   at 0 nT and a median average was used to estimate the transient correction while ignoring outlying values (Figure 4d).

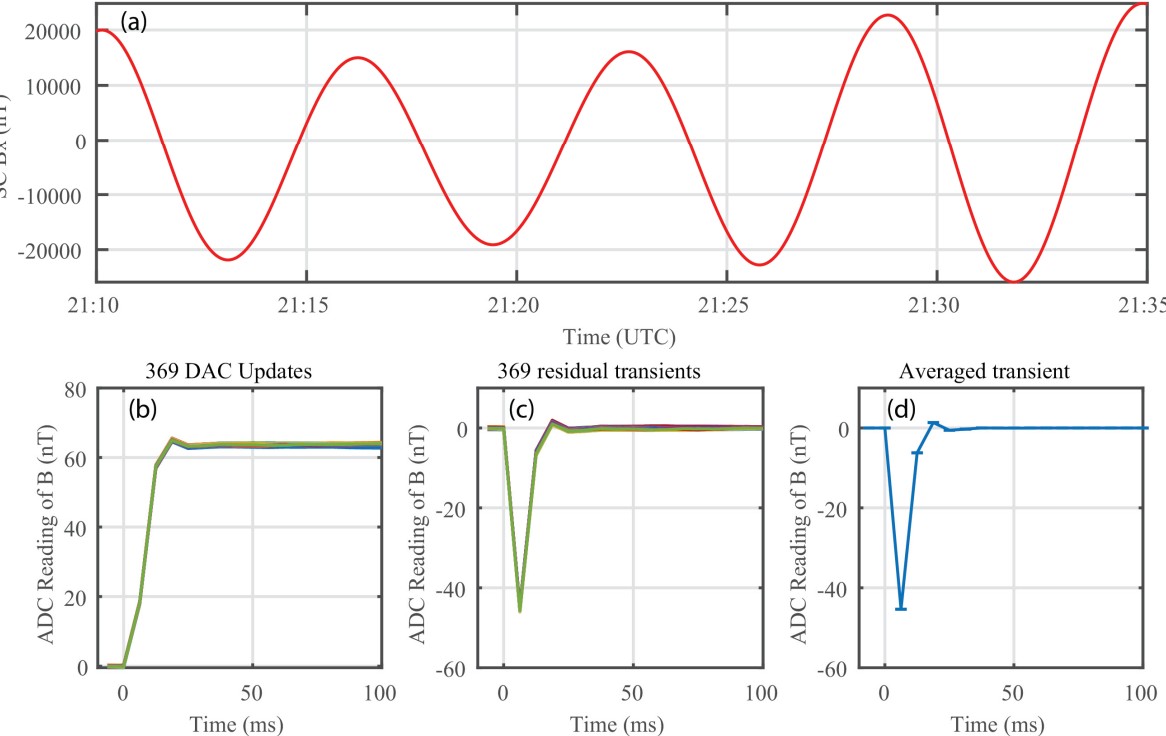

**Figure 4: (a) Timeseries of sensor X axis showing the spacecraft spin with no reaction wheels powered on. (b) Overplot of 369 ADC sequences following DAC updates. (c) Same but with the expected step from the DAC update removed to show the residual transient. (d) Median average estimate of the transient used to correct the measured data.**

15   Figure 5 shows the eighteen corrections fitted from the spinning no-wheel data corresponding to the instrumental X, Y, and Z axes on both the inboard and outboard MGF sensors. The ADC transient resulting from a DAC update on the same channel in the instrument (e.g., DACX on ADCX) were ~40-60 nT whereas the cross-channel transients (e.g., DAC X on ADC Y) were ~0.5-2.0 nT. Cross-channel updates due to DAC Z were larger than those resulting from DAC X or Y; this may be related to channel Z being constructed from the series connection of the sense windings around both ring cores in the sensor (Wallis et al., 2015).



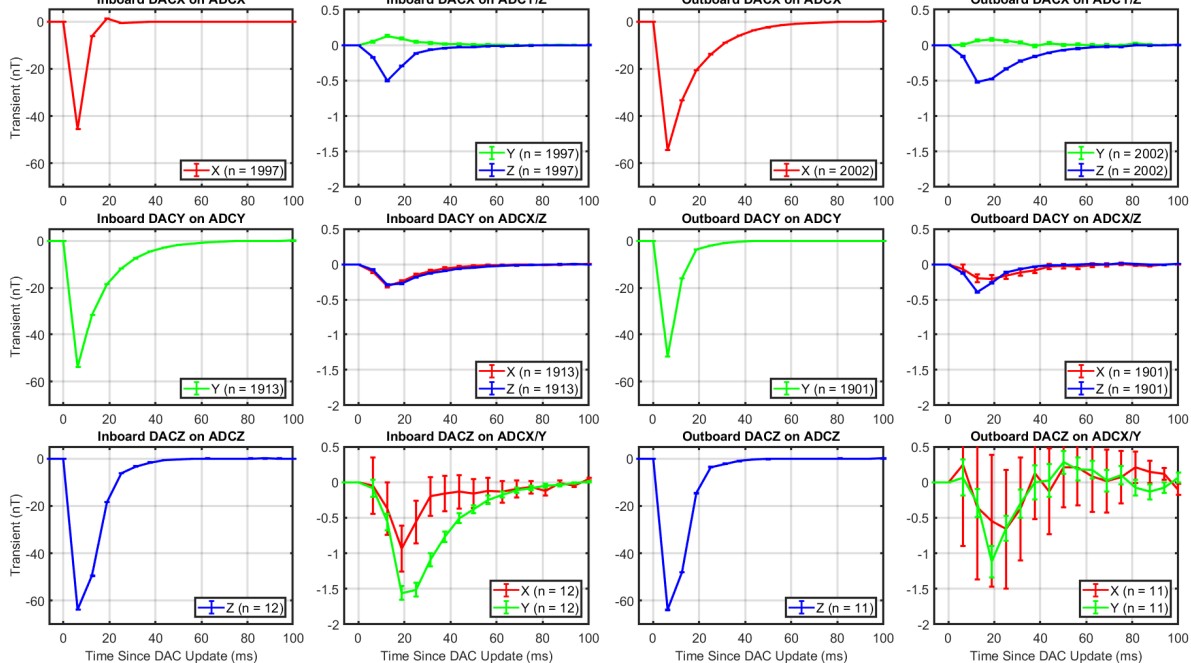

**Figure 5: Estimates of transient behaviour of each channel of both MGF instruments. The larger transients due to DAC Z may be related to Z being a series connection of two sense coils. The uncertainty in the Z Channel estimates results from fewer usable DAC updates as the spacecraft is primarily rotating around the instrument Z axis. The cross-channel transients are displayed on an expanded scale.**

Applying these fitted correction coefficients to several years of MGF observations showed that the transient behaviour was dependent on temperature. However, due to technical restrictions in the spacecraft recovery process, the safe-mode no-wheel data can only be obtained after the MGF had been in the shadow of the spacecraft for some time, causing the instrument to be unusually cold (-20 to -40 °C). Therefore, the fitted corrections are unsuitable for normal science operations that typically occur over a much wider range of warmer temperatures (up to +20 °C).

In 2016, one of Cassiope's original four reaction wheels failed. Rebalancing the spacecraft attitude control required that the remaining three wheels be slowed from ~15 Hz to ~1 Hz to stabilize the spacecraft attitude control system. This impacted the MGF magnetic data quality but does make the reaction wheels' spin speed slow enough that their magnetic signatures can be fitted and removed on the timescale of the transients. Transient fitting was repeated for all data taken after the wheels were slowed (June 2016 to December 2018). The data were then sorted into temperature bands from -30 to +20 °C and fitted as before to create temperature-dependent correction coefficients, shown in Figure 6.





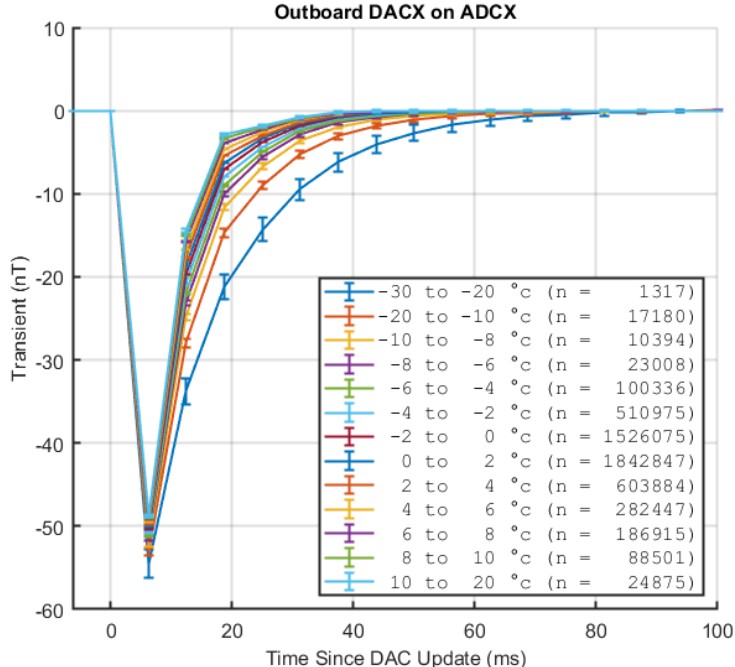

**Figure 6: The coefficients used to correct the transient following a DAC update in each instrument channel were found to be temperature dependent. "n" indicates the number of transients averaged in that temperature band.**

### 4 Results

5    Corrections for the transients in the MGF data are applied by subtracting the characterised ADC transients from the reported ADC readings after each DAC update. Figure 7 shows a representative correction for DAC ±2 steps showing the uncompensated (dashed) and compensated (solid) reconstructed magnetic field in the X, Y, and Z channels. The compensation suppresses the transient behaviour by a factor of at least 100 in each case.



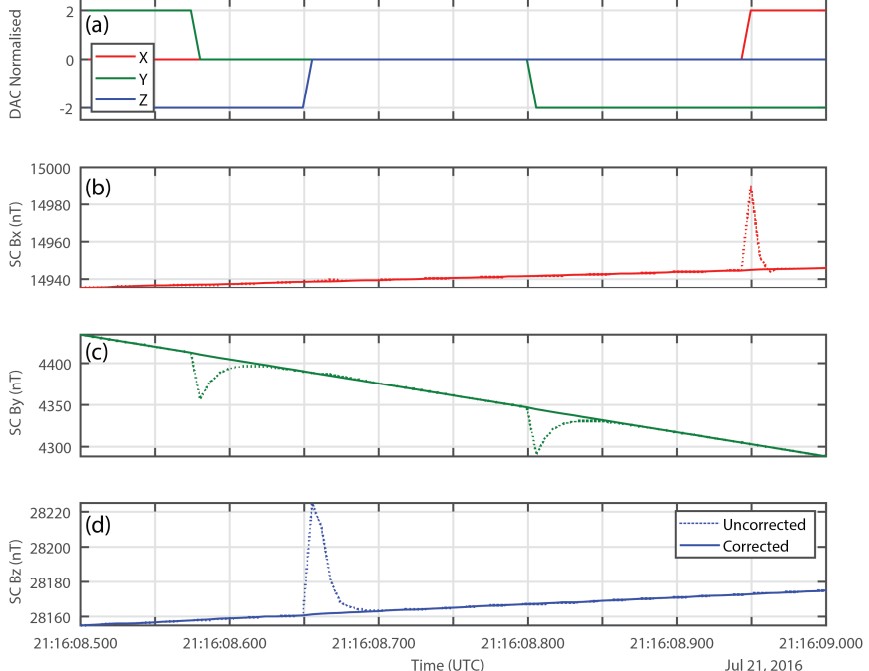

**Figure 7: Uncorrected and corrected time series following DAC updates (a) for the X (b), Y (c), and Z (d) instrument axes.**

Figure 1 uses a dynamic spectrum to visualise the spectral content of the uncompensated (a) and compensated (b) timeseries. Note that the amplitude of the vertical striping, caused by the broad-band content of the transients, is almost completely suppressed.

5   Figure 8 shows an event studied in Shen et al., (2018) where e-POP encountered large in-situ field variations associated with ion heating and downflow that require instrumental slew rates exceeding 3000 nT/s. These large-amplitude, short-duration magnetic field variations cause near-continuous stepping of the digital magnetic feedback to track the field. Consequently, feedback transients affect most of the data (Figure 8). The first attempt to mitigate the instrument transients (MGF data processing software version 1.0.0 and above) invalidated all affected samples and then interpolated (Figure 8, dashed pink). The reconstruction

10   described here (MGF data processing software version 2.0.0, Figure 8, solid green) overlaps the previous reconstruction when the underlying data is not affected by DAC update transients (Figure 8, black). However, it is evident that the interpolated version can miss entire structures of ~100 nT and 100 ms duration in regions of high instrument slew rates.

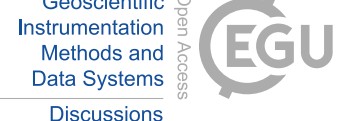



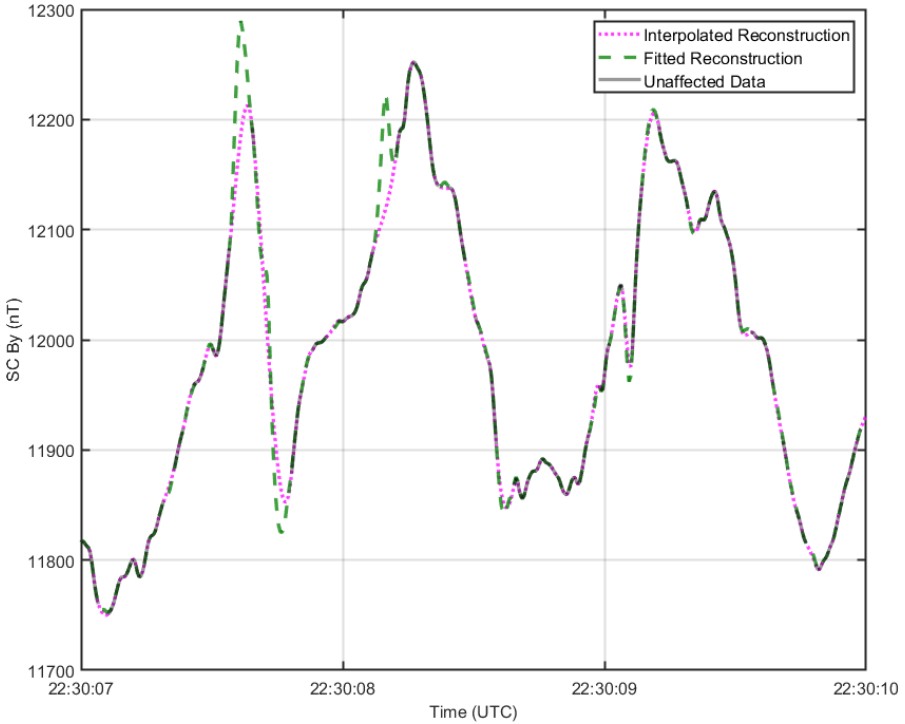

**Figure 8: Cross-track magnetic field measurements of an ion heating event studied by Shen et al. (2018) with high-amplitude small-scale magnetic perturbations. The presented transient compensation technique captures ~100 nT and ~200 ms features which are lost completely when the data is reconstructed by dropping data affected by DAC updates and interpolating.**

## 5 Conclusions and Future Work

The described in-situ characterisation and compensation successfully mitigates the transients in the MGF data following updates to the digital magnetic feedback. The current compensation is based on 2.5 years of data and the transients have been fit to better than 0.5 nT for most instrument temperatures. This correction allows MGF to resolve large-amplitude and small-scale magnetic features of ~100 nT and <1 km which were missed entirely by previous data interpolation methods. Ongoing data collection is

10 expected to allow us to continue to refine these correction coefficients providing cleaner and more useful spectral plots. More generally, this characterisation process should apply to other situations where the dynamics of an offsetting instrument must be calibrated in-situ.

### Code Availability

The MGF data processing software (mgftools) described herein is maintained in the Cassiope mission Subversion repository.

15 Release versions of this code are available at: https://epop.phys.ucalgary.ca/data/

### Data Availability

Science data from the e-POP mission is available via http via an open date-driven folder tree at: https://epop-data.phys.ucalgary.ca/

e-POP data can also be accessed using the e-POP Data Explorer (eDEx) tool available at: https://epop.phys.ucalgary.ca/data/#edex



**Author Contributions**

D. M. Miles is the PI for the MGF payload on e-POP, developed the data product processing code for the MGF, performed the in-situ calibration described here, and prepared the manuscript with contributions from all co-authors. A. D. Howarth developed the ground data-product processing software infrastructure and helped test the V2.0.0 data processing code. G. A. Enno is the e-POP

mission manager and coordinated the special safe-mode operations that produced the wheel-free data used in this manuscript.

**Competing Interests**

The authors declare that they have no conflict of interest.

**Acknowledgements**

Work on the project and the operation of the e-POP payload on the Cassiope Satellite is funded by the European Space Agency

through the Third Party Mission Programme as Swarm-Echo, a member of the Swarm Constellation. Work on the project was also supported by the Canadian Space Agency. D. M. Miles was supported by faculty start-up funding from the University of Iowa. The authors wish to thank D.D. Wallis, B.B. Narod, J.R Bennest and J.E. Schmidt for insight into the operation of the MGF instrument. A.W. Yau, M.J. Miles, S.E. Miles, and R.M. Broadfoot provided comments on an early copy of this manuscript.

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
