# Peer review of "In-situ Calibration of Offsetting Magnetometer Feedback Transients on the Cassiope Spacecraft"

_Geoscientific Instrumentation, Methods and Data Systems, 2019_

## Referee Comment (RC1) · Anonymous Referee #1 · 3 May 2019

General comments The paper concerns an on-orbit calibration process of the offsetting fluxgate magnetometer on the Cassiope spacecraft. The developed characterisation process for fluxgate magnetometer feedback dynamics allows a significant magnetic interference reduction from the reaction wheels and improvement of the instrument's magnetic field resolution for short time intervals ($\sim$100 ms). The paper is useful for on-orbit refinement of the calibrating process of an offsetting fluxgate magnetometer. Specific comments P. 2, Fig. 1 should be better explained: 1. Whether all broadband noise in Fig. 1a (vertical lines in STFT spectrum for time interval 7:00 . . . 7:50) was eliminated (cf. Fig. 1b) in the same way as for time interval 7:31:22 . . . 7: 31:23 (Figs. 1c-1d)? 2. Whether the signals near frequencies 17, 34, 51 Hz for time interval 7:00 . . . 7:50 are the 1st, 2nd and 3rd harmonics of magnetic interference from the reaction

wheels? 3. In which way the strong interference signal ≈40 Hz in time intervals ≈7:00 ... 7:10 and ≈7:47 ... 7:50 (Fig. 1a) was eliminated (cf. Fig. 1b)? P. 7, Fig. 6: The curves in temperature range -10 ... +10 oC are almost indistinguishable for the usual page format. So, it is desirable to show these curves in two subfigures, for example in different time scales (or in log time scale). Technical corrections P. 4, Fig. 3: The a), b), c) symbols should be added to appropriate subfigures.

---

## Referee Comment (RC2) · Anonymous Referee #2 · 17 May 2019

**Contents**

[Figure]

**1 General**

- The paper investigates on a scheme to in-flight fix a bug in the dynamics of the sensors in vector field experiment MGF (ePOP project) on the satellite `CASSIOPE`. The in-flight re-calibration problem occurs after an update in the MGF sensor operations to fix a timing bug uncovered after launch. The described basic laboratory solution to adopt the dynamic behaviour was applicable at the spacecraft only during special conditions. The paper reports on the achievements and shortcomings.

- Due to the significant inference by the attitude controlling wheels, the on-site determination of the required correction of the dynamic behaviour was, unfortunately, initially possible only on special occasions: when the satellite is running in save mode for a while, without the attitude control wheels operating. Some of this save-mode periods during recovery without wheels in operation were used to fit new corrections to the misbehaving transients.

- Ironically, just the fail of one of the attitude control wheels changed their impact on the MGF readings and subsequently allows to apply the correction of the transients for a wider span of temperature bins. The temperature span available was the major drawback on the application of the transient correction to all data beforehand.

- Even the constrictions of the applied method narrows the applicability, the description may well fit into the scope of the journal and may be useful to know for the community, in particular for users of ePOP MGF data.
**2 Detailed comments:**

- Figure 1 and it's descriptions, page 2, need a bit more information:

  – The earlier of both 'orange circle'-marks do not show something **very** disturbed to me, at least not in the printed version I'm looking at. This may be worth an additional short note to highlight the different characteristics.

  – What is that fairly monochromatic signal of the *De-trended Bx* we are looking at in frames 'c' and 'd'?

  – What is causing the varying amplitudes of the vertical strips? Is it the same effect as it's shown in Figure 3, where the characteristic of the irregular ticks are changing with some systematic as well. Seems to depend on sign and slope. Please add a short explanation.

- Timing, together with Figure 3, page 4:

  – Mentioned in the label is the 'engineering spare' magnetometer, but shown are data, presumably, for only one (arbitrary) **sensor** component. Otherwise the unspecified B may suggest to be the total field.

  – As the 'transient' behaviour is the main focus of the paper, some more early general information about the design decisions of the MGF 'offsetting' layout are missing: as the (fairly high) sampling rate and what is triggering the updates in detail. That the updates are not given on a fix time raster is deducible by the last sentence of page 4.

  – That the transient behaviour is stable in a slowly and linear drifting laboratory offset is shown in the top panel of figure 3, but is it also repeatable in nonlinear, i.e. in turbulent environments or with more erratic fields from field aligned currents (FACs) at polar regions? That may be important, if the

ePOP MGF data may be used in calculating Small Scale FACs or such (as the example of figure 8 may claim. . . ).

The piece wise linearity seems to be important, as on page 5, line 5, it is stated there: 'The sinusoidal trend created by the spacecraft spin can be approximated as linear over the 32 sample (200 ms) interval'. Please add a comment.

- Timing, Figure 4, page 5:

  – Are that horizontal ticks on the plotted line in figure `4d` error-bars? That may be obvious from Figure 5, but should be stated here as well.

- The major improvement after the fail of one attitude control wheel was possible because of (page 6, line 15): 'wheel's spin speed slow enough that their magnetic signatures can be fitted and removed on the timescale of the transients'. This finally crucial, now possible preprocessing step is neither described in a bit more detail nor supported by an illustrative example.

- To Figure 6 and descriptions:

  – Where is the temperature measured? Inside or outside the magnetometer? Is it a temperature sensor placed on the magnetic field sensor or at the electronic box? I'm surprised, that temperature induces such a large impact on the transients at the sensor itself.

- To Figure 8:

  – For completeness: As the label states, that it's $B_y$ in S/C system – which sensor is shown?

- Code availability, page 9

  – The `mgftools` downloadable `zip`-archive file for Linux do contain a 'sav' file with 'compiled' `IDL` routines only, which gives no deep insight into the code itself.

  On the `ePOP` website the downloadable `mgftools\_v3.2.zip` is declared for **visualizing** level 0 data products and this promised functionality is matched. But that is not what the term `data processing software` (in the paper page 9 line 14) suggest to me.

**3 Summary:**

Some clarifications are recommended to increase the readability and lucidity of the scheme and it's application.

---

## Author Comment (AC1) · 17 Jul 2019

We thank the referee for the constructive comments which we have incorporated into the manuscript. Referee #2 raised several questions and issues which we address below; the referee's comments are in plain text our responses in *italics* and any content added to or changed in the manuscript are in *"quoted italics*

General comments:

The paper concerns an on-orbit calibration process of the offsetting fluxgate magnetometer on the Cassiope spacecraft. The developed characterisation process for fluxgate magnetometer feedback dynamics allows a significant magnetic interference reduction from the reaction wheels and improvement of the instrument's magnetic field resolution for short time intervals (~100 ms). The paper is useful for on-orbit refinement of the calibrating process of an offsetting fluxgate magnetometer.

Specific comments:

P. 2, Fig. 1 should be better explained:

1. Whether all broadband noise in Fig. 1a (vertical lines in STFT spectrum for time interval 7:00 . . . 7:50) was eliminated (cf. Fig. 1b) in the same way as for time interval 7:31:22 . . . 7: 31:23 (Figs. 1c-1d)?

*The broadband noise in Fig. 1a results from the magnetic feedback transients and they are removed (Fig. 1b) using the techniques described in the manuscript. We have updated the figure caption to clarify this.*

*Change made – caption for Figure 1 now reads: "These transients introduce bursts of broadband noise that manifest as vertical lines in the uncompensated spectra"*

2. Whether the signals near frequencies 17, 34, 51 Hz for time interval 7:00 . . . 7:50 are the 1st, 2nd and 3rd harmonics of magnetic interference from the reaction wheels?

*Yes – those signals are the reaction wheel frequencies and their harmonics. We have clarified this in the figure caption.*

*Change made – caption for Figure 1 now reads: "The magnetic signatures of the spacecraft reaction wheels and their harmonics are visible near 17, 34, and 51 Hz and broaden during spacecraft manoeuvres around 07:05 and 07:50 UTC"*

3. In which way the strong interference signal ≈40 Hz in time intervals ≈7:00 . . . 7:10 and ≈7:47 . . . 7:50 (Fig. 1a) was eliminated (cf. Fig. 1b)?

*The spacecraft is maneuvering in those intervals causing many rapid updates to the magnetic feedback and the resulting transients. We have added additional clarifying text immediately before Figure 1.*

*Text added: "The spacecraft manoeuvres around 07:05 and 07:50 UTC resulting in the visible spectral widening of the reaction wheel tones and their harmonics as the various wheel rates change. All components of the magnetometer experience rapid change as the spacecraft rotates requiring many rapid updates to the digital feedback and resulting in the strong interference signal observed in the uncompensated spectra in Figure 1a. Characterizing and correction these transients, as described herein, significantly mitigates this effect resulting in the cleaner spectra shown as Figure 1b."*

P. 7, Fig. 6: The curves in temperature range -10 . . . +10°C are almost indistinguishable for the usual page format. So, it is desirable to show these curves in two subfigures, for example in different time scales (or in log time scale).

*Figure 6 has been updated as requested. The original plot is shown as (a) and a second panel (b) has been added showing 5-25 ms. The shorter timespan more clearly shows the curves for -10 . . . +10°C.*

*The caption how reads: "Figure 6: (a) The coefficients used to correct the transient following a DAC update in each instrument channel were found to be temperature dependent. "n" indicates the number of transients averaged in that temperature band. (b) Expanded time-scale to show ordering of the temperature dependent ticks."*

Technical corrections

P. 4, Fig. 3: The a), b), c) symbols should be added to appropriate subfigures.

*Change made.*

---

## Author Comment (AC2) · 17 Jul 2019

We thank the referee for the constructive comments which we have incorporated into the manuscript. Referee #2 raised several questions and issues which we address below; the referee's comments are in plain text our responses in *italics* and any content added to or changed in the manuscript are in *"quoted italics*

1 General

• The paper investigates on a scheme to in-flight fix a bug in the dynamics of the sensors in vector field experiment MGF (ePOP project) on the satellite CASSIOPE. The in-flight re-calibration problem occurs after an update in the MGF sensor operations to fix a timing bug uncovered after launch. The described basic laboratory solution to adopt the dynamic behaviour was applicable at the spacecraft only during special conditions. The paper reports on the achievements and shortcomings.

• Due to the significant inference by the attitude controlling wheels, the on-site determination of the required correction of the dynamic behaviour was, unfortunately, initially possible only on special occasions: when the satellite is running in save mode for a while, without the attitude control wheels operating. Some of this save-mode periods during recovery without wheels in operation were used to fit new corrections to the misbehaving transients.

• Ironically, just the fail of one of the attitude control wheels changed their impact on the MGF readings and subsequently allows to apply the correction of the transients for a wider span of temperature bins. The temperature span available was the major drawback on the application of the transient correction to all data beforehand.

• Even the constrictions of the applied method narrows the applicability, the description may well fit into the scope of the journal and may be useful to know for the community, in particular for users of ePOP MGF data.

2 Detailed comments:

• Figure 1 and it's descriptions, page 2, need a bit more information:

– The earlier of both 'orange circle'-marks do not show something very disturbed to me, at least not in the printed version I'm looking at. This may be worth an additional short note to highlight the different characteristics.

*We have added additional text to describe the two transients that are shown and that the larger of the transients is clipped in the figure by the y-scale required to resolve the smaller transient.*

*Text added: "Two examples are highlighted in **Error! Reference source not found.**c by orange circles. The first is a < 1 nT and results from a feedback step cross-coupled from another channel. The second is ~50*

*nT (clipped by the y-scale required to show the first transient) resulting from a feedback step on the same channel."*

*The caption for Figure 1 now reads "… The orange circles show two uncompensated feedback transients – one < 1 nT and one ~50 nT."*

– What is that fairly monochromatic signal of the De-trended Bx we are looking at in frames 'c' and 'd'?

*The multiple reaction wheels used to control the attitude of the spacecraft create the enveloped quasi-sinusoidal signal apparent in 'c' and 'd'. Clarifying text has been added to the caption for Figure 1.*

*Text added: "The magnetic signatures of the spacecraft reaction wheels and their harmonics are visible near 17, 34, and 51 Hz and broaden during spacecraft manoeuvres around 07:05 and 07:50 UTC. In the time domain (c,d) the reaction wheels create the visible enveloped sinusoidal signal."*

– What is causing the varying amplitudes of the vertical strips? Is it the same effect as it's shown in Figure 3, where the characteristic of the irregular ticks are changing with some systematic as well. Seems to depend on sign and slope. Please add a short explanation.

*The data for Figure 3 is taken after the firmware update so the transients are quite predictable. The different intensities of the vertical stripes relate to how often the magnetic feedback is updated. When the field changes rapidly in a component the feedback is updated more frequently leading to a higher power spectral density. Explanatory text has been added to the manuscript:*

*Text added: "The varying intensity of the vertical stripes primarily relates to how often feedback is updated. Frequent updates aggregate to a higher power spectral density."*

• Timing, together with Figure 3, page 4:

– Mentioned in the label is the 'engineering spare' magnetometer, but shown are data, presumably, for only one (arbitrary) sensor component. Otherwise the unspecified B may suggest to be the total field.

*Figure 3 shows data for the Sensor Bx component. The axes labels of the figure have been updated accordingly.*

*Change made: Figure 3 y-axis label now reads "Sensor Bx (nT)"*

– As the 'transient' behaviour is the main focus of the paper, some more early general information about the design decisions of the MGF 'offsetting' layout are missing: as the (fairly high) sampling rate and what is triggering the updates in detail. That the updates are not given on a fix time raster is deducible by the last sentence of page 4.

*We have added the relevant details immediately before Figure 2.*

*Text added: "*A proportional control loop updates every four samples (40 Hz) and varies the magnetic feedback to hold the field in the sensor near-zero. Feedback on each axis is controlled independently and triggers if the residual field exceeds 32, 64, or 128 nT and is stepped in those same increments back towards zero. Most feedback updates are 32 nT – the higher levels being triggered occasionally rapid field changes such as during auroral crossings.*"*

– That the transient behaviour is stable in a slowly and linear drifting laboratory offset is shown in the top panel of figure 3, but is it also repeatable in nonlinear, i.e. in turbulent environments or with more erratic fields from field aligned currents (FACs) at polar regions? That may be important, if the ePOP MGF data may be used in calculating Small Scale FACs or such (as the example of figure 8 may claim. . . ).

*The reviewers raise an important point here which is difficult for us to test with our current experimental setup. However, several two factors argue that the transient behavior is relatively stable. Firstly, the transients captured in the laboratory did not appear to vary with slew rate of the triangle wave used to exercise the engineering spare instrument. Secondly, the transients resulting from the three different feedback step sizes appear the same except scaled to the size of the feedback step (although our data here is thin as the instrument almost exclusively uses the smallest feedback step size).*

*Text added: "The transients did not appear to depend on the slew-rate of the field used to exercise the engineering sensor and scale with the size of the feedback step. This suggests the characterized transients should be repeatable even in the more erratic environmental fields experienced on orbit, although this is difficult verify experimentally."*

The piece wise linearity seems to be important, as on page 5, line 5, it is stated there: 'The sinusoidal trend created by the spacecraft spin can be approximated as linear over the 32 sample (200 ms) interval'. Please add a comment.

*The ~5-minute spin period produces shows very little curvature on the 200 ms timescale of the transient. The reaction wheels were more challenging to remove, and we have added additional description of that process in response to another question raised below by the reviewer.*

• Timing, Figure 4, page 5:

– Are that horizontal ticks on the plotted line in figure 4d error-bars? That may be obvious from Figure 5, but should be stated here as well.

*Yes – the ticks in 4d are error bars. Explanatory text has been added to the caption:*

*Change made – text for the caption of Figure 3 now reads: "(d) Median average estimate of the transient used to correct the measured data. Error bars show the variance of the averaged ADC sequences."*

• The major improvement after the fail of one attitude control wheel was possible because of (page 6, line 15): 'wheel's spin speed slow enough that their magnetic signatures can be fitted and removed on the timescale of the transients'. This finally crucial, now possible preprocessing step is neither described in a bit more detail nor supported by an illustrative example.

*We have added an illustrative example (new Figure 6) and text describing how this fitting was completed.*

*Text added: "Figure 1c shows an over plot of 617 ADC sequences from the inboard Bx channel following updates to the magnetic feedback DAC on the same channel. The background trend of the local magnetic field has been removed by subtracting a robust linear regression that excludes the region affected by the transient. Although the scatter is visibly broader than for the no-wheel data shown in* **Error! Reference source not found.** *the trend of the transient is clear and can be extracted by median filtering as shown in Figure 1d."*

*New Figure 6 added.*

[Figure]

*Figure 1: Equivalent analysis to **Error! Reference source not found.** but for an interval with slowed reaction wheels. (a) Timeseries of sensor X axis showing essentially one orbit of data. (b) Overplot of 617 ADC sequences following DAC updates. (c) Same but with the expected step from the DAC update removed to show the residual transient. (d) Median average estimate of the transient used to correct the measured data. Error bars show the variance of the averaged ADC sequences.*

• To Figure 6 and descriptions:

– Where is the temperature measured? Inside or outside the magnetometer? Is it a temperature sensor placed on the magnetic field sensor or at the electronic box? I'm surprised, that temperature induces such a large impact on the transients at the sensor itself.

*The temperatures described in the manuscript was measured by the sensor mounted on each instrument's electronics card. The temperature of the sensors were recorded as well but did not have a significant effect on the transients.*

*Text added: The text describing Figure 6 now reads "… dependent on temperature of the instruments' electronics package (the sensor temperatures were recorded as well but did not have a significant effect)."*

• To Figure 8:

– For completeness: As the label states, that it's By in S/C system – which sensor is shown?

*The data shown is from the Inboard Sensor. The y-axis label for Figure 8 now reads "Inboard SC By (nT)"*

• Code availability, page 9

– The mgftools downloadable zip-archive file for Linux do contain a 'sav' file with 'compiled' IDL routines only, which gives no deep insight into the code itself.

On the ePOP website the downloadable mgftools_v3.2.zip is declared for visualizing level 0 data products and this promised functionality is matched. But that is not what the term data processing oftware (in the paper page 9 line 14) suggest to me.

*The reviewer is correct – the data-processing software used to create this manuscript was not being correctly propagated to the e-POP website. It is now listed as "mgftools_matlab_source_alpha1.zip"*

3 Summary:

Some clarifications are recommended to increase the readability and lucidity of the scheme and it's application.